# Toxicological Impacts and Mechanistic Insights of Bisphenol a on Clear Cell Renal Cell Carcinoma Progression: A Network Toxicology, Machine Learning and Molecular Docking Study

**DOI:** 10.3390/biomedicines13112778

**Published:** 2025-11-13

**Authors:** Jie Chen, Biao Ran, Bo Chen, Jingxing Bai, Shibo Jian, Yin Huang, Jiahao Yang, Jinze Li, Zeyu Chen, Qiang Wei, Jianzhong Ai, Liangren Liu, Dehong Cao

**Affiliations:** Department of Urology, Institute of Urology, West China Hospital, Sichuan University, 37 Guoxue Lane, Chengdu 610041, China; chenjie@stu.scu.edu.cn (J.C.); ranbiao@stu.scu.edu.cn (B.R.); boychen@wchscu.edu.cn (B.C.); 2023224020109@stu.scu.edu.cn (J.B.); cheneyjie@foxmail.com (S.J.); hy201508@outlook.com (Y.H.); 2022224025422@stu.scu.edu.cn (J.Y.); lijinze@alu.scu.edu.cn (J.L.); doctorczy@wchscu.edu.cn (Z.C.); weiqiang@scu.edu.cn (Q.W.); jianzhong.ai@scu.edu.cn (J.A.)

**Keywords:** bisphenol A, renal cell carcinoma, machine learning, network toxicology, molecular docking

## Abstract

**Background:** Clear cell renal cell carcinoma (ccRCC) is a prevalent urological malignancy, accounting for approximately 1.6% of all cancer-related deaths in 2022. While endocrine-disrupting chemicals (EDCs) have been implicated as risk factors for ccRCC, the toxicological profiles and immune mechanisms underlying Bisphenol A (BPA) exposure in ccRCC progression remain inadequately understood. **Materials and Methods:** Protein–protein interaction (PPI) analysis and visualization were performed on overlapping genes between ccRCC and BPA exposure. This was followed by Gene Ontology (GO) and Kyoto Encyclopedia of Genes and Genomes (KEGG) enrichment analyses to elucidate potential underlying mechanisms. Subsequently, 108 distinct machine learning algorithm combinations were evaluated to identify the optimal predictive model. An integrated CoxBoost and Ridge regression model was constructed to develop a prognostic signature, the performance of which was rigorously validated across two independent external datasets. Finally, molecular docking analyses were employed to investigate interactions between key genes and BPA. **Results:** A total of 114 overlapping targets associated with both ccRCC and BPA were identified. GO and KEGG analyses revealed enrichment in cancer-related pathways, including pathways in cancer, endocrine resistance, PD-L1 expression and PD-1 checkpoint signaling, T-cell receptor signaling, endocrine function, and immune responses. Machine learning algorithm selection identified the combined CoxBoost-Ridge approach as the optimal predictive model (achieving a training set concordance index (C-index) of 0.77). This model identified eight key genes (*CHRM3*, *GABBR1*, *CCR4*, *KCNN4*, *PRKCE*, *CYP2C9*, *HPGD*, *FASN*), which were the top-ranked by coefficient magnitude in the prognostic model. The prognostic signature demonstrated robust predictive performance in two independent external validation cohorts (C-index = 0.74 in cBioPortal; C-index = 0.81 in E-MTAB-1980). Furthermore, molecular docking analyses predicted strong binding affinities between BPA and these key targets (Vina scores all <−6.5 kcal/mol), suggesting a potential mechanism through which BPA may modulate their activity to promote renal carcinogenesis. Collectively, These findings suggested potential molecular mechanisms that may underpin BPA-induced ccRCC progression, generating hypotheses for future experimental validation. **Conclusions:** These findings enhance our understanding of the molecular mechanisms by which BPA induces ccRCC and highlight potential targets for therapeutic intervention, particularly in endocrine and immune-related pathways. This underscores the need for collaborative efforts to mitigate the impact of environmental toxins like BPA on public health.

## 1. Introduction

Kidney cancer is a significant global health concern, recognized as an immunogenic tumor with rising incidence and mortality rates [1]. In 2022, approximately 434,419 new cases and 155,728 deaths were reported worldwide, ranking it 14th in incidence and 16th in mortality among all cancers [2]. Clear cell renal cell carcinoma (ccRCC) is the most common type, accounting for approximately 85% of all kidney cancers [3]. Standard treatments include surgical resection—partial or radical nephrectomy—and targeted therapies such as tyrosine kinase inhibitors (e.g., sunitinib). Despite therapeutic advancements, many patients continue to experience disease progression and poor post-treatment outcomes [4]. Beyond health implications, kidney cancer imposes substantial economic burdens. A U.S. study projected that the total cost of cancer care would increase by 27%, from $124.57 billion in 2010 to $157.77 billion by 2020, primarily due to demographic changes [5]. Globally, the economic cost of kidney cancer reached billions of dollars in 2020, with further increases anticipated in the coming decades [6]. These trends underscore the urgent need for research into the underlying causes of ccRCC and the development of more effective, cost-efficient treatments.

Environmental factors are critical contributors to cancer development, with increasing evidence implicating widespread industrial pollutants as significant risk factors [7]. Among these pollutants, Bisphenol A (BPA; 2,2-bis(4-hydroxyphenyl) propane) is particularly concerning due to its extensive use in polycarbonate plastics and epoxy resins. BPA is found in numerous consumer products—such as food containers, toys, medical devices, and electronics—leading to pervasive human exposure [8]. As an endocrine-disrupting chemical (EDC), BPA structurally resembles synthetic estrogen, enabling it to bind estrogen receptors and exert both estrogenic and anti-estrogenic effects, thereby disrupting endocrine function. These disruptions are linked to impaired reproductive health and early puberty [9]. Additionally, epidemiological and animal studies have associated BPA exposure with an increased risk of cancers, including leukemia, ovarian cancer, and breast cancer [10,11,12]. According to the International Agency for Research on Cancer (IARC), BPA is classified as Group 3 carcinogenicity [13]. In response to growing health concerns, many countries have implemented regulatory measures to limit BPA exposure. For instance, the European Union introduced stringent regulations in 2018, restricting BPA migration in food contact materials to below 0.05 mg/kg and banning its use in polycarbonate baby bottles and infant drinking containers [14].

Previous studies have frequently detected BPA in blood and urine samples, highlighting pervasive human exposure [15]. EDCs like BPA, along with genetic predispositions and other environmental factors, have been implicated in the occurrence and progression of urological malignancies. For example, Tse et al., provided epidemiological evidence of BPA’s carcinogenic effects on the human prostate [16], while Pellerin et al. demonstrated that urinary BPA impairs normal urinary tract cells and promotes bladder cancer cell growth [17]. Similarly, ccRCC development has been linked to environmental exposures such as heavy metals and air pollution, as well as immune-modulating gene mutations and chronic inflammation [18,19]. Despite advancements in understanding urological cancers, the relationship between BPA exposure and ccRCC remains poorly understood, and the broader impact of environmental pollutants on ccRCC progression necessitates further exploration.

Network toxicology integrates bioinformatics, big data analytics, and genomics to explore toxic pathways and disease mechanisms [20]. By employing network-based approaches, researchers can map complex interactions linking chemicals, biological targets, and adverse outcomes [21]. Molecular docking, a computational method for predicting ligand–target interactions, advances drug discovery by simulating how small molecules bind to biological macromolecules [22]. Integrated studies utilizing molecular docking have revealed how toxins interact with biological molecules, highlighting their mechanisms of toxicity. For instance, molecular docking has been used to study aflatoxin B1-induced hepatotoxicity and Bis(2-ethylhexyl) phthalate (DEHP)-related prostatic carcinogenesis [23,24]. These examples underscore the effectiveness of combining network toxicology with molecular docking to uncover disease mechanisms and guide targeted therapies.

In this study, we employ an integrated approach that combines network toxicology and molecular docking to identify the molecular targets and mechanisms by which BPA contributes to renal carcinogenesis and ccRCC progression. By elucidating these interactions, we aim to provide insights that may inform the development of more effective preventive and therapeutic strategies for ccRCC.

## 2. Methods

The software, online databases, and tools utilized in this study are listed in Appendix A.

### 2.1. Preliminary Network Analysis of BPA Toxicity

To evaluate the potential toxicity of Bisphenol A, we performed an initial assessment using computational prediction tools, supplemented by an extensive literature review. Toxicological data were obtained from three widely recognized online platforms: ADMETlab 2.0 (https://admetmesh.scbdd.com/service/evaluation/cal (accessed on 13 February 2025)), ProTox 3.0 (https://tox.charite.de/protox3/ (accessed on 13 February 2025)), and VNN-ADMET (https://vnnadmet.bhsai.org/vnnadmet/ (accessed on 13 February 2025)). These platforms provided predictive insights into the endocrine-disrupting and carcinogenic potential of BPA. By integrating results from these computational tools with relevant literature findings, we developed a comprehensive understanding of BPA’s toxicity profile. This multifaceted approach facilitated the identification of key mechanisms involved in BPA-induced toxicity, elucidating its potential contribution to disease progression.

### 2.2. Construction of BPA Targets Library

The canonical SMILES notation and molecular structure of BPA were retrieved from the PubChem database (https://pubchem.ncbi.nlm.nih.gov/ (accessed on 13 February 2025)). Utilizing this molecular information, we identified potential toxicity targets through cross-database searches using SwissTargetPrediction (http://www.swisstargetprediction.ch/ (accessed on 13 February 2025)), STITCH (http://stitch.embl.de/ (accessed on 13 February 2025)), and ChEMBL (https://www.ebi.ac.uk/chembl/ (accessed on 13 February 2025)). Searches were restricted to human-related targets to ensure biological relevance. After removing duplicates, we generated a refined and comprehensive list of potential BPA-related toxicity targets.

### 2.3. Selection of ccRCC-Related Target Network

To identify targets associated with renal cell carcinoma, we used the keywords “Kidney cancer,” “Renal cancer,” and “Clear cell renal cell carcinoma” in three disease-specific databases: GeneCards (https://www.genecards.org/ (accessed on 13 February 2025)), OMIM (https://omim.org/ (accessed on 13 February 2025)), and TTD (https://db.idrblab.net/ttd/ (accessed on 13 February 2025)). The retrieved targets were consolidated, and duplicates were eliminated. We applied a median “score” threshold to ensure a strong association with ccRCC; only genes with scores above the median were included in the final ccRCC disease target library.

### 2.4. Screening of Key Targets and Construction of Protein–Protein Interaction Network

An intersection analysis of BPA-related and ccRCC-related targets was conducted to identify potential targets involved in BPA-induced renal carcinogenesis. Venn diagrams were generated using the Venny 2.1 tool (https://bioinfogp.cnb.csic.es/tools/venny/index.html (accessed on 13 February 2025)) to visualize overlapping targets. These intersecting targets were imported into the STRING database (http://string-db.org/ (accessed on 13 February 2025)) to construct a protein–protein interaction (PPI) network, with a minimum interaction score threshold of 0.4. The network was visualized using Cytoscape 3.9.1, and key targets were identified using the MCODE plug-in for clustering and the CytoNCA plug-in for centrality analysis, highlighting the most influential nodes based on connectivity.

### 2.5. Gene Function and Pathway Enrichment Analysis of Target Proteins

To elucidate the biological roles of the identified targets in BPA-induced ccRCC tumorigenesis, we performed Gene Ontology (GO) and Kyoto Encyclopedia of Genes and Genomes (KEGG) pathway enrichment analyses using the DAVID database (https://davidbioinformatics.nih.gov/ (accessed on 14 February 2025)). GO analysis provided functional annotations across three categories: biological processes (BP), cellular components (CC), and molecular functions (MF). Pathways with a false discovery rate (FDR) less than 0.05 were considered statistically significant. The top 10 GO terms and top 20 KEGG pathways were selected for further investigation. Additional analyses were conducted using FUMA (https://fuma.ctglab.nl/gene2func/ (accessed on 14 February 2025)) and Metascape (https://metascape.org/ (accessed on 14 February 2025)) to gain deeper insights into the involved signaling pathways. Visualization and interpretation of the GO and KEGG enrichment results were facilitated using the WeiShengXin tool (https://www.bioinformatics.com.cn/ (accessed on 14 February 2025)), enabling exploration of toxicant–disease signaling pathways and biological processes, and highlighting potential toxicities and mechanistic insights.

### 2.6. Development of a BPA-ccRCC Prediction Model Using Machine Learning

To establish a robust prognostic model with high predictive accuracy, we adapted the methodology described by Liu et al. [25]. Utilizing the Mime1 R package (v0.0.0.9000), we employed the TCGA ccRCC dataset (KIRC) as the training cohort. Independent external validation cohorts were derived from: (i) the ccRCC dataset published in Nature 2013 [26] (accessed via cBioPortal), and (ii) the E-MTAB-1980 dataset [27] (available through ArrayExpress). Subsequently, ten distinct machine learning algorithms were applied to the BPA-ccRCC-associated genes: Random Survival Forests (RSF), Elastic Net (Enet), Lasso regression, Ridge regression, Stepwise Cox regression, CoxBoost, Cox partial least squares regression (plsRcox), Supervised Principal Components (SuperPC), Generalized Boosted Regression Models (GBM), and Survival Support Vector Machines (survival-SVM). The concordance index (C-index) was computed for each algorithm combination. Combinations were systematically ranked based on their performance, and the model achieving the highest mean C-index was selected as the optimal predictive model. Following this selection, an integrated CoxBoost and Ridge regression model was trained using the KIRC to develop the final prognostic signature. To quantify accuracy beyond discrimination, we report censoring-aware loss functions at clinically relevant horizons (1, 3, and 4.7 years). To rigorously validate the model’s predictive performance, it was tested on the two independent external validation cohorts. Based on calculated risk scores, patients were stratified into low-risk and high-risk groups. Statistical analyses were performed using R packages survival (v3.3.1) and survminer (v0.4.9) to compute Hazard ratios (HRs) with corresponding 95% confidence intervals (CIs), Log-rank *p*-values and Kaplan–Meier survival curves. Additionally, time-dependent area under the curve (AUC) values were calculated for each validation cohort to quantify the model’s predictive accuracy.

### 2.7. Prognostic Model Comparative Analysis

To benchmark the performance of our optimal model against established methodologies, we systematically acquired four previously published renal cell carcinoma (ccRCC) prognostic signatures from independent studies [28,29,30,31]. The constituent genes and their corresponding coefficients of each published model were retrieved. The risk scores for patients within each of our three cohorts (TCGA training set, cBioPortal validation cohort, and E-MTAB-1980 validation cohort) were recalculated using the reported coefficients. The predictive performance of these published models was then rigorously compared to that of our CoxBoost-Ridge integrated signature across all three cohorts using the C-index and AUC.

### 2.8. Tumor Microenvironment (TME) Characterization

Patients within the TCGA ccRCC cohort were stratified into low-risk and high-risk groups based on the prognostic scores derived from the CoxBoost-Ridge model. To investigate potential biological correlates of the prognostic signature, we characterized the tumor microenvironment using two complementary computational deconvolution approaches. The IOBR R package: Employed to infer TME cell abundances and functional features. The immunedeconv R package (v2.1.0): Utilized to estimate immune cell infiltration levels using multiple established algorithms (e.g., CIBERSORT, xCell, EPIC, MCP-counter). Correlation analyses were performed between the prognostic risk scores and the immune cell composition and functional state features of the TME derived from these deconvolution methods.

### 2.9. Molecular Docking of BPA with Key Targets

To elucidate the molecular interactions and binding modes between BPA and key target proteins, molecular docking simulations were conducted. BPA compound files in SDF format were obtained from the PubChem database. The three-dimensional structures of the top six target proteins were retrieved from AlphaFold2 (https://alphafold.ebi.ac.uk/ (accessed on 5 March 2025)), UniProt (https://www.uniprot.org/ (accessed on 5 March 2025)), and the Protein Data Bank (PDB) (https://www.rcsb.org/ (accessed on 5 March 2025)). Docking simulations were performed using the CB-Dock platform (https://cadd.labshare.cn/cb-dock2/ (accessed on 5 March 2025)), and the resulting binding poses were analyzed and visualized using Discovery Studio software (v2019).

## 3. Result

### 3.1. Basic In Silico Toxicity Profiling of BPA

Integrating outputs from predictive toxicity software tools yielded a comprehensive toxicity profile for BPA. The toxicity models indicated active toxicity endpoints correlated with carcinogenicity and identified the blood–brain barrier (BBB) as the associated membrane transporter affected by BPA. Consistent with previous literature, our model demonstrated that BPA-induced toxicity perturbs estrogen receptor alpha (ERα), the estrogen receptor ligand-binding domain (ER-LBD), and mitochondrial membrane potential (MMP) (Appendix A). These findings provide a foundation for systematic and in-depth exploration of BPA’s adverse effects on the human urinary system.

### 3.2. Acquisition of BPA-Elicited Clear Cell Renal Cell Carcinoma (ccRCC) Toxicity Targets

We identified 192 BPA-related toxicity targets from the SwissTargetPrediction, ChEMBL, and STITCH databases, and 5954 targets strongly associated with ccRCC through analysis of the GeneCards, OMIM, and TTD databases. Integrating these target sets and removing duplicates resulted in 114 overlapping targets as potential candidates mediating BPA-induced ccRCC (Figure 1). It presents a Venn diagram illustrating the intersection between BPA-related targets and ccRCC disease targets. Specifically, 192 BPA-related targets intersect with 5954 ccRCC-related targets, identifying 114 potential toxicity targets that may mediate BPA-induced ccRCC. A detailed list of these targets is provided in Appendix A.

### 3.3. PPI Network Analysis and Key Target Screening

A PPI network was constructed using the STRING database, consisting of 114 nodes and 1030 edges. The network exhibited an average node degree of 18.1, an average local clustering coefficient of 0.565, and a PPI enrichment *p*-value of <1.0 × 10^−16^, indicating statistically significant interactions among the identified targets. Topological properties of the network nodes, including degree and betweenness centrality, were analyzed using Cytoscape software, resulting in an optimized visualization of the PPI network (Figure 2). Figure 2 shows the PPI network of 114 potential targets, visually representing interactions between these targets. Node size and color correspond to degree values, while edge thickness and color reflect interaction strength.

### 3.4. Functional Annotation and Pathway Enrichment Analysis

GO and KEGG pathway analyses were conducted on the 114 potential ccRCC pathogenesis targets associated with BPA toxicity using the DAVID database, limited to Homo sapiens. A total of 599 statistically significant terms were identified in the GO analysis, including 412 biological processes, 66 cellular components, and 121 molecular functions. To prioritize the most relevant GO terms, we ranked them according to their FDR values and selected the top 10 terms with the lowest FDR for visualization in the enrichment analysis plot (Figure 3).

KEGG pathway analysis revealed 161 enriched signaling pathways, visualized as bubble plots and classification histograms (Figure 4), arranged in ascending order based on their FDR values. Both GO and KEGG analyses indicated that the identified genes are widely distributed across various subcellular localizations. Notably, many of these genes are involved in key regulatory processes such as neurotransmission, cell proliferation, cell cycle regulation, apoptosis, and signal transduction (Appendix A). Among the KEGG pathway enrichment results, several pathways emerged as particularly prominent, including “Pathways in Cancer”, “Endocrine Resistance”, “Lipid and Atherosclerosis”, “PD-L1 Expression and PD-1 Checkpoint Pathway in Cancer”, “T Cell Receptor Signaling Pathway”, “Chemical Carcinogenesis–Receptor Activation”, and “B Cell Receptor Signaling Pathway” (Appendix A). These pathways are closely associated with endocrine disruption and immune regulation. These findings suggest that BPA-induced ccRCC toxicity may be mediated through multiple biological processes and signaling pathways affecting both the immune and endocrine systems, providing insights into the potential molecular mechanisms underlying BPA’s carcinogenic effects in ccRCC.

### 3.5. Development and Validation of the BPA-ccRCC Prognostic Model

#### 3.5.1. Model Construction and Optimization

We systematically evaluated 108 combinatorial machine learning models derived from 10 distinct algorithms (Random Survival Forests, Elastic Net, Lasso, Ridge, Stepwise Cox, CoxBoost, plsRcox, SuperPC, GBM, survival-SVM). The CoxBoost + Ridge integrated model demonstrated superior predictive performance, achieving the highest mean concordance index (C-index = 0.773) across all combinations (Figure 5A). Subsequently, this optimal model was trained on the TCGA ccRCC cohort. The CoxBoost algorithm identified 23 key BPA-ccRCC-associated genes (Figure 5B). Ridge regression refined the model, with optimal generalization performance observed at log(λ) values between −2 and 0 (Figure 5C,D). The final prognostic signature retained all 23 genes (Appendix A), but the top eight genes by absolute coefficient magnitude (CHRM3, GABBR1, CCR4, KCNN4, PRKCE, CYP2C9, HPGD, FASN) were considered the most influential (Figure 5E). We evaluated censoring-aware loss metrics. The IPCW-Brier score increased with the prediction horizon (1y: 0.072; 3y: 0.134; 4.7y: 0.160) (Appendix A), while the IPCW-MAE was 0.152, 0.289, and 0.349 at 1, 3, and 4.7 years, respectively, indicating modest error at short-term horizons and larger error at longer horizons (Appendix A). The integrated Brier score (IBS) over the follow-up was 0.151, suggesting overall acceptable calibration or accuracy.

#### 3.5.2. Performance Validation

The signature significantly stratified patients into low- and high-risk groups with distinct survival outcomes (*p* < 0.001; Figure 5F) in internal validation (TCGA). Robust performance was confirmed in two independent cohorts: Nature^2013^ ccRCC cohort (cBioPortal; Figure 5G) and E-MTAB-1980 cohort (ArrayExpress; Figure 5H). The model maintained high predictive accuracy at 1-, 3-, and 4.7-year intervals in all cohorts (Figure 5I–K). The 4.7-year cutoff for E-MTAB-1980 was necessitated by its maximum follow-up (1740 days). Univariate Cox regression (via the Mime1 R package) confirmed the prognostic risk score as an independent hazard factor (HR, 5.97; 95% CI: 4.52 to 7.89; Figure 5L).

#### 3.5.3. Elevated Immune Infiltration in the High-Risk Subgroup

Stratification using the CoxBoost-Ridge signature revealed significantly elevated immune infiltration scores in high-risk patients within the TCGA cohort (Figure 6A), quantified using deconvolution algorithms (via IOBR and immunedeconv R packages).

#### 3.5.4. Comparison with the Established Prognostic Models

We retrieved four established ccRCC prognostic signatures from independent studies [28,29,30,31]. Coefficients and gene sets were applied to our cohorts to compute risk scores (Appendix A). Univariate Cox regression analyses were systematically performed across all datasets to evaluate the prognostic association of each model with patient outcomes. Crucially, the risk score derived from our CoxBoost-Ridge integrated signature demonstrated a significantly stronger association with poorer clinical outcomes compared to all other models (Figure 6B). Our CoxBoost-Ridge model achieved significantly higher C-indices than most published models across all cohorts (Figure 6C). The signature consistently ranked top for 1-, 3-, and 5-year AUCs in all validation cohorts (Figure 6D), demonstrating state-of-the-art prognostic capability.

### 3.6. Molecular Docking for BPA with Core Target of ccRCC

To explore the molecular mechanisms underlying BPA-induced ccRCC, we conducted detailed molecular docking simulations targeting the eight key proteins implicated in ccRCC pathogenesis: cholinergic receptor muscarinic 3 (CHRM3), gamma-aminobutyric acid type B receptor subunit 1 (GABBR1), C-C motif chemokine receptor 4 (CCR4), potassium calcium-activated channel subfamily N member 4(KCNN4), protein kinase C epsilon (PRKCE), cytochrome P450 family 2 subfamily C member 9 (CYP2C9), 15-hydroxyprostaglandin dehydrogenase (HPGD), and fatty acid synthase (FASN). These docking models were generated using the CB-Dock online tool, and all binding energies were below −6.5 kcal/mol, indicating strong binding affinities between BPA and these target proteins. This suggests that BPA can effectively bind to these core targets, potentially contributing to the molecular mechanisms of BPA-induced toxicity in ccRCC. Specific Vina scores and docking energies are provided in Table 1. The binding conformations with the lowest energy for each BPA–target complex were visualized using Discovery Studio software (v2019), producing both 2D and 3D structural representations (Figure 7)**.** These visualizations offer insights into the interaction dynamics between BPA and the identified targets, highlighting how BPA may interfere with key molecular pathways in ccRCC. The results underscore the potential role of these targets in mediating BPA-induced toxicity and advancing ccRCC pathogenesis.

## 4. Discussion

Despite advancements in diagnosis and management, renal cell carcinoma remains one of the most lethal urological malignancies [4]. Bisphenol A and its analogs, commonly used as plasticizers in various commercial and industrial products, are frequently detected in human biological samples such as urine, blood, and placenta, indicating widespread exposure and reproductive and endocrine toxicity [9]. While BPA has been associated with multiple cancers, including hepatocellular carcinoma and breast cancer [32], its effects on ccRCC have not been thoroughly investigated. Understanding how BPA influences ccRCC progression is crucial for developing effective prevention and treatment strategies.

In this study, using computational evaluation tools, we identified 114 potential targets related to BPA-induced ccRCC toxicity from the ChEMBL, STITCH, and GeneCards databases. By constructing an interaction network using the STRING database and Cytoscape, we identified 23 hub targets—including CHRM3, GABBR1, CCR4, KCNN4, PRKCE, CYP2C9, HPGD, and FASN—that are key mediators of BPA-induced renal carcinogenicity.

As a GPCR regulating smooth muscle contraction and glandular secretion, CHRM3 drives tumor progression in multiple cancers. It enhances prostate cancer invasiveness via Hippo–YAP axis reactivation under androgen deprivation and promotes glioblastoma cell motility [33,34], correlating with poor murine survival. Critically, its role in ccRCC remains unexplored, suggesting novel ligand-dependent oncogenic mechanisms. The gamma-aminobutyric acid type B receptor subunit 1 (GABBR1) is a metabotropic G protein-coupled receptor that encodes a γ-aminobutyric acid (GABA) receptor responsible for mediating GABAergic inhibitory neurotransmission. Dysfunction of GABBR1 has been implicated in the pathogenesis of neurological disorders such as schizophrenia and epilepsy. Beyond its neurological roles, GABBR1 plays a critical oncogenic role in colorectal [35] and prostate cancer [36]. This receptor promotes prostate cancer cell invasion by upregulating matrix metalloproteinase (MMP) expression. Our multi-omics analysis identifies GABBR1 as a top-ranked risk predictor (|coefficient| = 0.1719) in ccRCC, implying GABAergic signaling may subvert renal tissue homeostasis.

Elevated expression of the chemokine receptor CCR4 in tumors correlates with poor prognosis across multiple cancer types and has been extensively studied in ccRCC. Berlato et al. [37] detected abundant CCR4 expression in biopsy samples from advanced ccRCC patients and demonstrated that CCR4 antagonists remodeled the immune infiltration profile, resulting in reduced tumor weight in RCC-bearing mice. Potassium calcium-activated channel subfamily N member 4 (KCNN4), also known as KCa3.1, encodes a protein that forms part of a latent heterotetrameric voltage-insensitive potassium channel activated by intracellular calcium. Analogous to CCR4, the upregulated KCNN4 expression is associated with adverse prognosis in numerous malignancies, including papillary thyroid carcinoma [37], pancreatic cancer [38], and ccRCC [39]. KCNN4 modulates immune cells by increasing regulatory T cells (Tregs) while decreasing resting mast cells, thereby potentiating opportunities for immune evasion and diminishing the efficacy of immune checkpoint inhibitors [39].

PRKCE encodes Protein Kinase C-epsilon (PKCε), a calcium-independent isoform of the novel PKC subfamily that functions as a transforming oncogene. PKCε has been implicated in regulating cell invasion and motility [40], and contributes to oncogenesis across multiple malignancies, including breast cancer [41], non-small cell lung cancer (NSCLC) [42], and prostate cancer [43]. Basu et al. [44] demonstrated that PKCε promotes breast cancer cell survival not only by inhibiting apoptosis but also through induction of autophagy. Paradoxically, in ccRCC, while PRKCE mRNA expression is significantly downregulated [45], PKCε protein levels are elevated. This PKCε overexpression correlates with aggressive phenotypes in ccRCC [46]. Cytochrome P450 family 2 subfamily C member 9 (CYP2C9), located on chromosome 10, is detectable in multiple organs and exhibits aberrantly high expression in early-stage esophageal adenocarcinoma (EAC) [47]. CYP2C9 metabolizes diverse exogenous and endogenous compounds [48], notably catalyzing the conversion of arachidonic acid (AA) to epoxyeicosatrienoic acids (EETs) [49]. These EETs may promote cancer progression by altering the tumor microenvironment, thereby inducing cancer cell proliferation, survival, migration, and invasion. In summary, CHRM3, GABBR1, CCR4, KCNN4, PRKCE, and CYP2C9 are potential toxicological targets for BPA-induced ccRCC, providing a basis for future prevention and treatment strategies.

GO and KEGG pathway analyses revealed significant enrichment of BPA-related pathways in ccRCC development and progression, including immune, endocrine, and cancer-associated pathways. Immune-related pathways may be particularly important in BPA-induced ccRCC. Immune factors are critical components of the pathological microenvironment in ccRCC, and dysregulated immunity is closely linked to tumor progression and adverse outcomes [1,50]. Programmed cell death protein 1 (PD-1), an inhibitory receptor expressed on T cells, B cells, and myeloid cells, interacts with its ligand PD-L1 on tumor cells and antigen-presenting cells to downregulate immune responses, thereby maintaining immune homeostasis [51]. In ccRCC, PD-L1 expression is often upregulated, correlating with poor prognosis and increased tumor aggressiveness. The engagement of PD-1 by PD-L1 on tumor cells inhibits the anti-tumor immune response, allowing for unchecked tumor growth and progression. Inhibitors targeting these pathways are important immunotherapeutic agents for ccRCC [52,53]. This is consistent with our KEGG enrichment analysis findings, suggesting that BPA exposure may influence ccRCC occurrence through immune-related pathways such as the “PD-L1 Expression and PD-1 Checkpoint Pathway in Cancer,” “T Cell Receptor Signaling Pathway,” and “B Cell Receptor Signaling Pathway”.

Our findings indicate that the “Pathways in Cancer” signaling pathway significantly impacts BPA-induced renal toxicity. This pathway involves mechanisms such as signal transduction pathways (e.g., PI3K/AKT, mTOR, Ras/Raf/MEK/ERK) [54], angiogenesis via the VEGF signaling pathway, and genetic mutations including VHL and BAP1 [55]. Furthermore, the PI3K-AKT pathway plays a crucial role in the development and progression of ccRCC. The phosphatidylinositol-3-kinase (PI3K)/Akt and mammalian target of rapamycin (mTOR) signaling pathways are essential for many aspects of cell growth and survival in both physiological and pathological conditions [56]. This pathway is frequently disrupted in ccRCC, contributing significantly to tumor formation, disease progression, and therapeutic resistance [57]. Some drugs that inhibit the PI3K-AKT pathway have been used in the treatment of ccRCC [58,59].

Additionally, regarding direct toxic effects, BPA present in the bloodstream is metabolized and excreted by the kidneys, leading to its accumulation in renal tissues. Previous studies have found that BPA in urine can promote bladder tumorigenesis, suggesting potential carcinogenic effects on urinary tract tissues [60]. Therefore, we hypothesize that prolonged exposure of renal cells to BPA may promote carcinogenesis through “Chemical Carcinogenesis-Receptor Activation” pathways, potentially initiating or enhancing tumorigenic processes within the renal parenchyma. Early animal studies have also suggested potential nephrotoxic effects of plastic derivatives on renal function and the progression of chronic kidney disease (CKD). In murine models, administration of varying doses of BPA resulted in dose-dependent increases in serum creatinine and blood urea nitrogen levels, indicative of impaired renal function. Histopathological and electron microscopy examinations further confirmed glomerular damage [61]. Although a direct association between BPA exposure and increased incidence of ccRCC has not been established, the nephrotoxic effects of BPA may contribute to CKD progression. Given that CKD is a known risk factor for ccRCC and can complicate surgical and systemic management in ccRCC patients, the potential role of BPA in CKD progression may indirectly influence ccRCC incidence and outcomes. These observations underscore the necessity for further research to elucidate the mechanisms by which BPA and other plastic derivatives impact renal carcinogenesis and renal function, ultimately informing preventive and therapeutic strategies.

Similarly, we found that BPA may promote ccRCC occurrence and development through “Endocrine Resistance” and the MAPK signaling pathway, aligning with previous studies [62,63,64]. We also identified immune-related pathways such as “TH17 Cell Differentiation” and “IL-17 Signaling Pathway” that may be associated with BPA-induced ccRCC. Currently, research on these pathways is limited, and we hope that future studies will explore these associations further.

Molecular docking findings indicate that the environmental endocrine-disrupting chemical BPA specifically interacts with key targets essential for the function of renal cell carcinoma cells, including CHRM3, GABBR1, CCR4, KCNN4, PRKCE, CYP2C9, HPGD, and FASN. BPA can mimic or disrupt hormone functions and affect the immune system, which are critically important in the tumor immune microenvironment and the development and progression of hormone-dependent cancers [65]. The observed binding interactions involve hydrogen bonds and hydrophobic contacts, with specific bond distances playing a key role in stabilizing these ligand-protein complexes. These interactions highlight potential mechanisms by which BPA may contribute to the molecular etiology of ccRCC by altering key cellular pathways and functions. It is important to note that the molecular docking results presented here are computational predictions. While the strong binding affinities (Vina scores < −6.5 kcal/mol) are indicative of potential interactions, these findings require validation through experimental methods.

The European Union introduced stringent regulations in 2018, restricting BPA migration in food contact materials and banning its use in baby drinking containers [14]. Motivated by this need, we employed network toxicology and molecular docking to comprehensively map the complex molecular mechanisms of BPA-induced ccRCC toxicity. Network toxicology integrates advances in bioinformatics, genomics, and multi-level big data analysis, while molecular docking is a well-established in silico structure-based method suitable for predicting and simulating ligand–target interactions at the molecular level [20,21]. Our work enhances the efficiency, depth, and predictive accuracy of toxicological screening, revealing molecular mechanisms of diseases like BPA-induced ccRCC and enabling the prioritization of potential key targets associated with toxic phenotypes. By identifying these key molecular interactions and pathways, we provide valuable insights that could guide future research and therapeutic development. However, these results are derived solely from computational predictions (e.g., network toxicology, molecular docking), which are inherently hypothetical. The models and interactions identified require validation through wet-lab experiments, such as in vitro binding assays or in vivo models, to confirm causality and mechanistic roles.

## 5. Conclusions

In summary, this study identified 114 BPA-related pathogenic targets in ccRCC and highlighted 23 key targets—including AKT1, ESR1, PTGS2, and PPARG—that may play pivotal roles in BPA-induced renal carcinogenicity. By integrating network toxicology and molecular docking methods, we have deepened our understanding of the pathogenesis associated with environmental exposure in ccRCC and identified novel molecular targets that could facilitate the development of personalized diagnostic and therapeutic strategies.

Our work may ultimately improve outcomes for patients with urological malignancies by exploring potential clinical translational strategies. Collaboration among urologists, oncologists, environmental scientists, and industry experts is crucial to fully elucidating how plastic byproducts like BPA may impact the risk and progression of urological cancers. These findings support the notion that BPA, as an endocrine-disrupting chemical, has toxicological effects on ccRCC progression and provide a valuable basis for further research. However, additional toxicological and clinical studies are necessary to validate our conclusions.

## Figures and Tables

**Figure 1 biomedicines-13-02778-f001:**
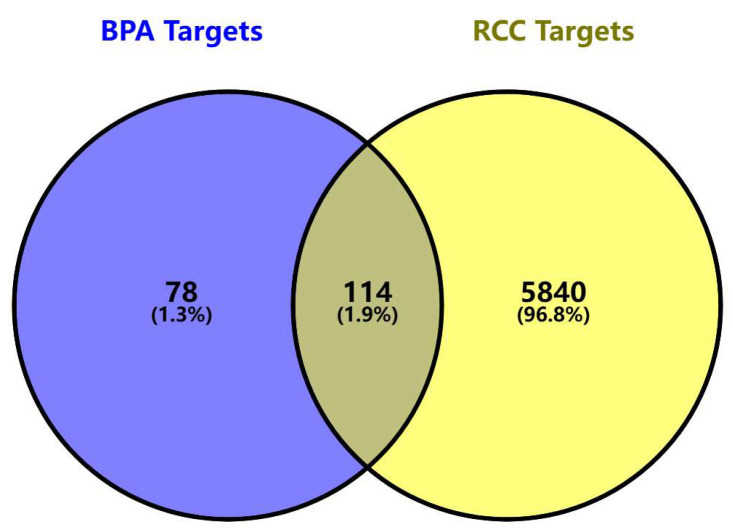
Venn diagram of the targets of BPA and ccRCC. The Venn diagram illustrates the intersection between BPA-related targets (blue) and renal cell carcinoma disease targets (yellow). The blue circle represents 192 BPA-related targets, which intersect with 5954 ccRCC-related targets, identifying 114 potential toxicity targets (grey region) that may mediate BPA-induced ccRCC progression.

**Figure 2 biomedicines-13-02778-f002:**
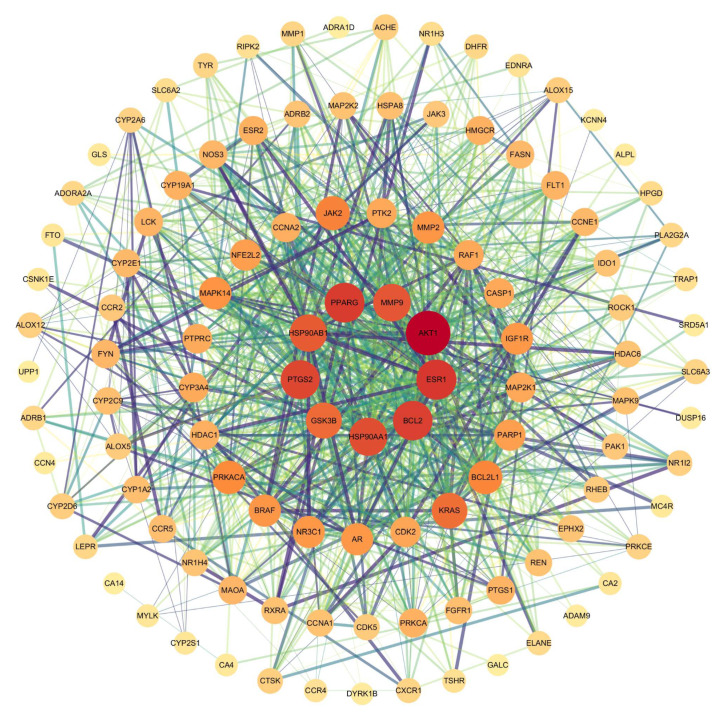
The PPI network of potential targets. The PPI network shows 114 potential targets, visually representing the interactions between these targets. Node size and color correspond to degree values (darker colors and larger sizes indicate higher degree), while edge thickness and color reflect interaction strength (thicker and darker edges indicate stronger interactions).

**Figure 3 biomedicines-13-02778-f003:**
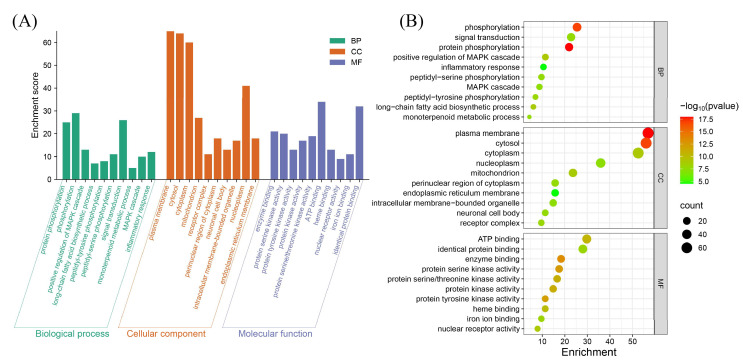
GO enrichment analysis for potential targets. (**A**) Histogram displaying the top 10 enriched terms in each GO category, ranked by FDR values. The height of each bar indicates the gene count, representing the degree of enrichment. (**B**) Bubble plot where bubble size reflects gene expression in a specific term, and color intensity corresponds to the FDR value—the smaller the FDR, the higher the enrichment significance.

**Figure 4 biomedicines-13-02778-f004:**
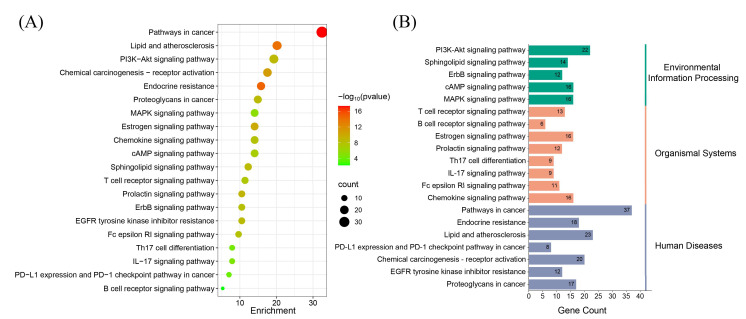
KEGG enrichment analysis of potential targets. (**A**) The bubble plot shows the top 20 enriched KEGG pathways, with bubble size representing the number of enriched genes and color intensity indicating pathway significance. (**B**) The histogram illustrates the enrichment frequency of each pathway, where bar length corresponds to the gene count, and color reflects the enrichment significance.

**Figure 5 biomedicines-13-02778-f005:**
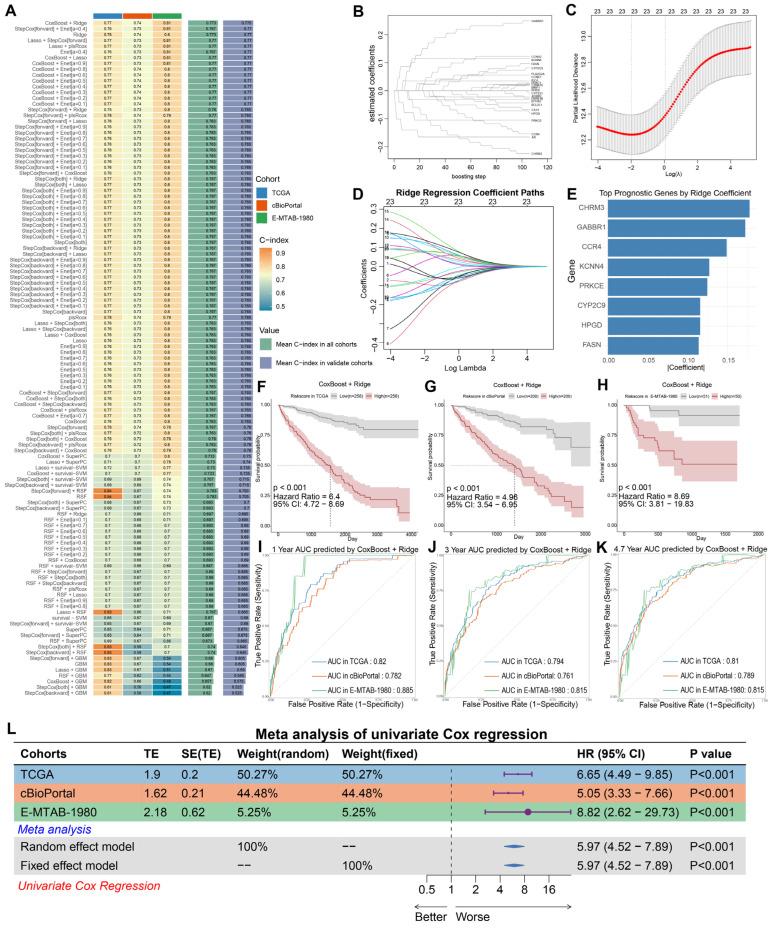
Construction and validation of the prognostic signature for BPA-associated clear cell renal cell carcinoma (ccRCC). (**A**) Comparative evaluation of 10 machine learning algorithms—including Random Survival Forests (RSF), Elastic Net (Enet), Lasso, Ridge, CoxBoost, plsRcox, Supervised Principal Components (SuperPC), Generalized Boosted Regression Models (GBM), and Survival Support Vector Machines (Survival-SVM)—based on concordance index (C-index) performance. (**B**) Identification of 23 key BPA-ccRCC-associated genes from 114 candidate targets using the CoxBoost algorithm. (**C**,**D**) Ridge regression optimization for prognostic gene selection in BPA-ccRCC model (Integrated with CoxBoost algorithm). (**C**) Regularization parameter tuning: Partial likelihood deviance curve (solid red line) versus Log(λ). The dashed vertical line marks optimal regularization strength where deviance stabilizes within a minimal confidence band (gray ribbon), balancing model complexity and generalizability. (**D**) Coefficient shrinkage paths: Trajectories of 23 CoxBoost-selected gene coefficients across Log(λ) values. Each colored line represents the coefficient shrinkage path of a single gene, with distinct colors used for visual discrimination among the 23 different genes. (**E**) Ranking of signature genes by absolute regression coefficient magnitude (CHRM3, GABBR1, CCR4, KCNN4, PRKCE, CYP2C9, HPGD, FASN shown). (**F**–**H**) Kaplan–Meier survival analysis stratified by risk score ((**F**) TCGA cohort, (**G**) cBioPortal validation cohort, (**H**) E-MTAB-1980 validation cohort). (**I**) Time-dependent ROC curves demonstrating the model’s predictive accuracy for 1-year survival (AUC: TCGA = 0.82; cBioPortal = 0.79; E-MTAB-1980 = 0.885). (**J**) Time-dependent ROC curves demonstrating the model’s predictive accuracy for 3-year survival (AUC: TCGA = 0.794; cBioPortal = 0.761; E-MTAB-1980 = 0.815). (**K**) Time-dependent ROC curves demonstrating the model’s predictive accuracy for 4.7-year survival (AUC: TCGA = 0.81; cBioPortal = 0.789; E-MTAB-1980 = 0.815). (**L**) Meta-analysis of univariate Cox regression validating the prognostic risk score as an independent hazard factor. Forest plot demonstrating pooled hazard ratio (HR) of 5.97 (95% CI: 4.52–7.89) for the CoxBoost-Ridge signature across three cohorts.

**Figure 6 biomedicines-13-02778-f006:**
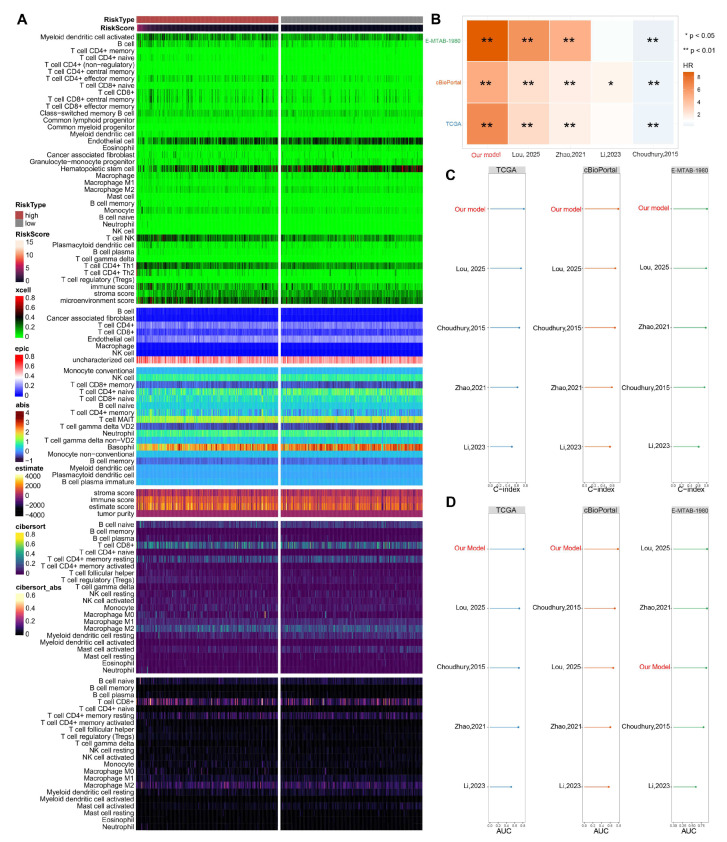
Comprehensive Characterization of Immune Infiltration and Benchmarking Against Established ccRCC Prognostic Models. (**A**) Heatmap matrices depicting risk score correlation with deconvoluted TME features using six computational methods (CIBERSORT, EPIC, xCell, MCP-counter Absolute Score, ESTIMATE, CIBERSORT Absolute Mode). (**B**) Hazard ratio comparison of the CoxBoost-Ridge prognostic model against established ccRCC signatures across three independent cohorts. CoxBoost-Ridge achieved significantly higher HR across all cohorts, indicating superior risk stratification. (**C**) C-index of the CoxBoost-Ridge combined model and 4 published models across 3 cohorts. (**D**) 1-year AUC of the CoxBoost-Ridge combined model and 4 published models across 3 cohorts [28,29,30,31].

**Figure 7 biomedicines-13-02778-f007:**
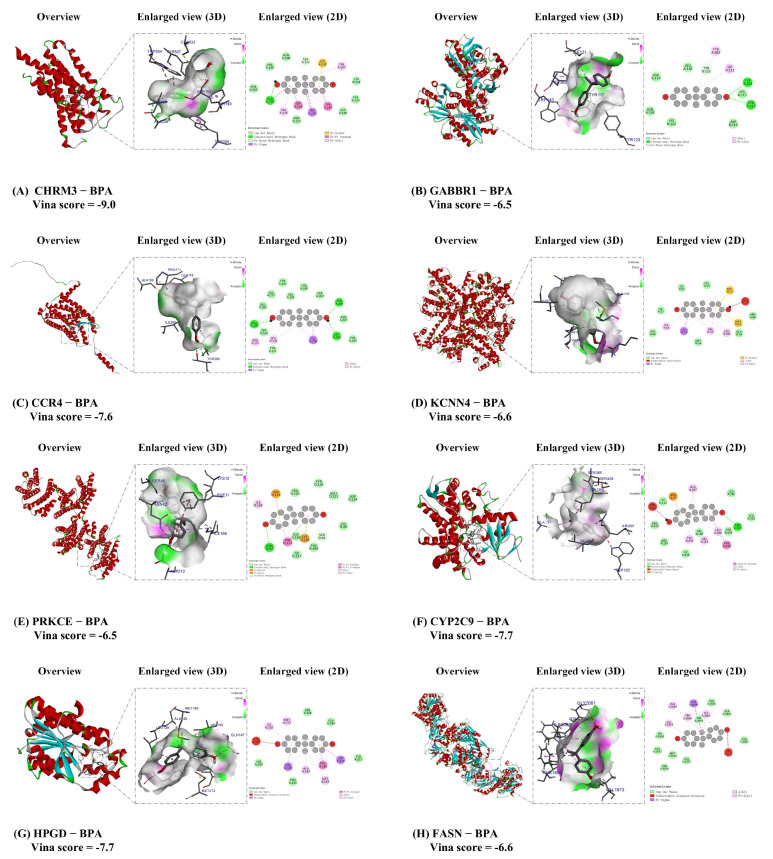
Molecular docking results of the lowest binding energy of the core target protein with the BPA. (**A**) CHRM3 and BPA; (**B**) GABBR1 and BPA; (**C**) CCCR4 and BPA; (**D**) KCNN4 and BPA. The red circle focuses on the details of the key sites where BPA binds to the KCNN4 protein, with the core being the ARG H:74 (arginine 74) residue; (**E**) PRKCE and BPA; (**F**) CYP2C9 and BPA. The red circle focuses on the details of the key sites where BPA binds to the CYP2C9 protein, with the core being the TRP A:120 (tryptophan 120) residue; (**G**) HPGD and BPA. The red circle focuses on the details of the key sites where BPA binds to the HPGD protein, with the core being the GLN A:147 (glutamine 147) residue; (**H**) FASN and BPA. The red circle focuses on the details of the key sites where BPA binds to the HPGD protein, with the core being the GLY A:2061 (Glycine 2061) residue. Each docking conformation is visualized using Discovery Studio software, showing both 2D and 3D structural representations.

**Table 1 biomedicines-13-02778-t001:** Molecular docking results of core target protein with the BPA. (BPA, CC(C)(C1=CC=C(C=C1)O)C2=CC=C(C=C2)O).

	Gene Name	Uniprot ID	PDB ID	Vina Score_min_ (Kcal/mol)	Vina Score (kcal/mol) [Median (min, max)] (from 5 Molecular Docking Runs)
1	*CHRM3*	P20309	8EA0	−9.0	−7.0 (−9.0, −5.2)
2	*GABBR1*	Q9UBS5	4MQE	−6.5	−6.2 (−6.5, −5.6)
3	*CCR4*	P51679	Selected by AlphaFold3	−7.6	−5.4 (−7.6, −5.3)
4	*KCNN4*	O15554	9OA8	−6.6	−6.2 (−6.6, −5.8)
5	*PRKCE*	Q02156	2WH0	−6.5	−5.9 (−6.5, −5.7)
6	*CYP2C9*	P11712	5A5I	−7.7	−7.4 (−7.7, −6.3)
7	*HPGD*	P15428	2gdz	−7.7	−5.6 (−7.7, −5.3)
8	*FASN*	P49327	8VF7	−6.6	−6.6 (−7.5, −6.6)

## Data Availability

Data are available from the corresponding author if justification for the requirement is justified.

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
