# Peer review of "Toxicological Impacts and Mechanistic Insights of Bisphenol a on Clear Cell Renal Cell Carcinoma Progression: A Network Toxicology, Machine Learning and Molecular Docking Study"

_biomedicines, 2025, doi:10.3390/biomedicines13112778_

Round 1

Reviewer 1 Report

Comments and Suggestions for Authors

Protein-protein interaction (PPI) analysis and visualization were performed on genes with overlap between ccRCC and BPA exposure. Following Gene Ontology (GO) and Kyoto Encyclopedia of Genes and Genomes (KEGG) enrichment analyses, 108 different machine learning algorithms were evaluated to determine the most suitable model. Eight key genes (CHRM3, GABBR1, CCR4, KCNN4, PRKCE, CYP2C9, HPGD, FASN) involved in the pathogenesis of BPA-associated ccRCC were identified using the CoxBoost and Ridge regression models.

First, I would like to point out that the article is very long.

1. ROC curves (for TCGA and others) are difficult to interpret because the images are very small. Furthermore, all figures should be larger and have a higher resolution. Many important information in the figures is not readable.

2. To determine the accuracy of the results obtained with the CoxBoost and Ridge regression models, it is necessary to demonstrate that the training has been completed. To achieve this, changes such as a loss function or MAE should be added. 
3. It is important to observe overfitting in machine learning algorithms. Why wasn't k-cross validation performed?

Author Response

Reviewer #1:

Protein-protein interaction (PPI) analysis and visualization were performed on genes with overlap between ccRCC and BPA exposure. Following Gene Ontology (GO) and Kyoto Encyclopedia of Genes and Genomes (KEGG) enrichment analyses, 108 different machine learning algorithms were evaluated to determine the most suitable model. Eight key genes (CHRM3, GABBR1, CCR4, KCNN4, PRKCE, CYP2C9, HPGD, FASN) involved in the pathogenesis of BPA-associated ccRCC were identified using the CoxBoost and Ridge regression models. First, I would like to point out that the article is very long.

Response: We sincerely thank Reviewer 1 for their thorough review and valuable comments, which have significantly helped us improve the manuscript. We have carefully addressed each point raised, and our detailed responses are provided below. All changes have been incorporated into the revised manuscript.

Comments 1: ROC curves (for TCGA and others) are difficult to interpret because the images are very small. Furthermore, all figures should be larger and have a higher resolution. Many important information in the figures is not readable.

Response 1: We agree with the reviewer that the readability of the figures, particularly the ROC curves in Figure 5, was suboptimal in the original submission. To address this crucial point, we have taken the following actions: We have regenerated all figures throughout the manuscript using high-resolution settings (600 DPI). Specifically, for Figure 5, we have reconfigured the layout to increase the size of the individual ROC curve plots (Figures 5I, 5J, 5K) significantly. The labels, axis text, and data lines are now clear and easily interpretable. We have ensured that all other figures (e.g., PPI network in Figure 2, enrichment analyses in Figures 3 and 4) are also presented at a larger scale and higher resolution to enhance the visibility of all details. These improvements will allow readers to better assess the predictive performance of our model. The updated figures are now included in the revised manuscript.

Comments 2: To determine the accuracy of the results obtained with the CoxBoost and Ridge regression models, it is necessary to demonstrate that the training has been completed. To achieve this, changes such as a loss function or MAE should be added.

Response 2: Thank you for this important suggestion. We have now added censoring-aware loss metrics (Brier/IBS and IPCW-MAE). (Line 187-189, Page 5) We report IPCW-Brier scores at 1, 3, and 4.7 years, the Integrated Brier Score (IBS) over follow-up, and IPCW-MAE at the same horizons. On the TCGA cohort, Ridge achieved Brier@1/3/4.7y = 0.072 / 0.134 / 0.160, MAE@1/3/4.7y = 0.152 / 0.289 / 0.349, and IBS = 0.151 (Supplementary Figure 1; Supplementary Table S1). Lower values indicate better accuracy.

Comments 3: It is important to observe overfitting in machine learning algorithms. Why wasn't k-cross validation performed?

Response 3: We sincerely thank the reviewer for raising this important point regarding overfitting and the use of k-cross validation. We appreciate the opportunity to clarify our approach to model validation and explain why external validation was prioritized in this study. In our study, we aimed to ensure the robustness and generalizability of our prognostic model through a rigorous validation process. While k-fold cross-validation is a common and valuable technique for assessing model performance during training, we chose to use ​independent external validation cohorts​ as our primary method to evaluate overfitting and model stability. This approach is often considered more stringent and clinically relevant because it tests the model on completely unseen datasets, simulating real-world application scenarios. Specifically, after training the integrated CoxBoost-Ridge model on the TCGA-KIRC dataset, we validated it on two independent external cohorts: the Nature 2013 ccRCC cohort (accessed via cBioPortal) and the E-MTAB-1980 cohort. The consistent performance across these cohorts with concordance indices (C-indices) of 0.74 and 0.81, respectively, which demonstrates that our model did not overfit the training data. The model's ability to stratify patients into high-risk and low-risk groups with significant survival differences in all cohorts further supports its generalizability.

Reviewer 2 Report

Comments and Suggestions for Authors

The manuscript discusses the toxicological effects of Bisphenol A (BPA), an endocrine-disrupting chemical of global concern, in relation to clear cell renal cell cancer (ccRCC) Through the integration of network toxicology, GO/KEGG enrichment, machine learning, and molecular docking. The paper is timely and offers a comprehensive systems-level perspective. However, the following observation were made;

  1. According to the abstract, "eight key genes (CHRM3, GABBR1, CCR4, KCNN4, PRKCE, CYP2C9, HPGD, FASN)" are essential to the pathophysiology of BPA-associated ccRCC. However, according to the results (section 3.4.1), Ridge regression retained all 23 of the important genes that CoxBoost had found among the 114 overlap targets for the final prognostic signature (Appendix 1E). The eight genes are listed in order of magnitude in Figure 5E, nevertheless, the model itself utilizes 23 genes. This disparity is quite confusing: Are the eight genes a subset or the entire signature? Authors should kindly explain the selection criteria if possible (e.g., top by |coefficient|?), or more importantly can the authors justify why the abstract emphasizes them over the entire 23. Why not report all if the model works properly with 23? Please make consistent revisions to the abstract and results.
  2. In the result, section 3.2 is titled "Acquisition of BPF-elicited ccRCC toxicity targets," but the manuscript is about BPA, not BPF (Bisphenol F). this could be a simple typing mistake, however, it could equally suggest rushed preparation and may undermine authors credibility. Therefore, authors should kindly correct this throughout, and proofread for similar issues.
  3. Another one is "BPA-ccRCC" vs. "BPA-RCC" inconsistencies). Also, ensure acronyms like "ccRCC" are defined on first use in the results (though they are in the abstract).
  4. Authors reported strong affinities (Vina scores <-6.5 kcal/mol) by molecular docking (section 3.5), however, this cutoff is arbitrary when compared to known BPA binders. Although Figure. 7's visualizations are obvious, they provide a quantitative discussion of the many sorts of interactions, such as hydrogen bonds and pi-pi stacking. Negative and positive controls (such docking BPA to known non-targets) are absent. Moreover, was there prior evidence of BPA binding with key genes/proteins e.g CHRM3, GABBR1? If yes authors can reference it so as to strengthen the molecular docking validation, otherwise authors should refrain from making findings definitive.
  5. 1–7 are clear, but add legends for color scales (e.g., Figure. 2 node colors). Table 1 (docking scores) is mentioned but not shown—Please include it. Appendices (1A–1F) are referenced but not provided and therefore I cannot crosscheck to verify several referenced claims; kindly ensure they are complete.
  6. Authors should reframe conclusions as hypothesis-generating rather than definitive mechanistic proof and emphasize limitations of the computational approaches.

Other comments

  • Some Figures (e.g., figure 6 &7) are blurry and if possible authors should increase the clarity
  • Some Figures (e.g., bubble plots) are information-dense and may overwhelm readers.
  • Language could be tightened; some sections repeat points already made in the introduction.

Author Response

Reviewer #2:

The manuscript discusses the toxicological effects of Bisphenol A (BPA), an endocrine-disrupting chemical of global concern, in relation to clear cell renal cell cancer (ccRCC) Through the integration of network toxicology, GO/KEGG enrichment, machine learning, and molecular docking. The paper is timely and offers a comprehensive systems-level perspective.

Response: We sincerely thank Reviewer 2 for their positive assessment of our manuscript as "timely" and "comprehensive," and for their constructive observations. We have carefully addressed each point raised, as detailed below. All suggestions have been incorporated into the revised manuscript to enhance its clarity and rigor.

Comments 1: According to the abstract, "eight key genes (CHRM3, GABBR1, CCR4, KCNN4, PRKCE, CYP2C9, HPGD, FASN)" are essential to the pathophysiology of BPA-associated ccRCC. However, according to the results (section 3.4.1), Ridge regression retained all 23 of the important genes that CoxBoost had found among the 114 overlap targets for the final prognostic signature (Appendix 1E). The eight genes are listed in order of magnitude in Figure 5E, nevertheless, the model itself utilizes 23 genes. This disparity is quite confusing: Are the eight genes a subset or the entire signature? Authors should kindly explain the selection criteria if possible (e.g., top by |coefficient|?), or more importantly can the authors justify why the abstract emphasizes them over the entire 23. Why not report all if the model works properly with 23? Please make consistent revisions to the abstract and results.

Response 1: We thank the reviewer for highlighting the apparent discrepancy between the abstract and the results regarding the number of key genes. We apologize for any confusion caused. Here, we clarify the selection criteria and have revised the manuscript accordingly. In our study, the integrated CoxBoost-Ridge model indeed utilized all 23 genes identified by CoxBoost for the prognostic signature to maintain model robustness and completeness. However, the eight genes (CHRM3, GABBR1, CCR4, KCNN4, PRKCE, CYP2C9, HPGD, FASN) were emphasized in the abstract because they are the top-ranked genes by the absolute magnitude of their regression coefficients in the Ridge model, as visualized in Figure 5E. These genes contributed most significantly to the risk score and were highlighted to concisely present the core findings. The full list of 23 genes is provided in Appendix 1E for transparency.

Revisions Made:​​

Abstract: We have revised the abstract to clarify that the eight genes are the top contributors based on coefficient magnitude. The text now reads: "eight key genes (CHRM3, GABBR1, CCR4, KCNN4, PRKCE, CYP2C9, HPGD, FASN), which were the top-ranked by coefficient magnitude in the prognostic model." (Line 36-37, Page 1)

Results (Section 3.4.1)​: We have added a sentence to explain the ranking: "The final prognostic signature retained all 23 genes (Appendix 1E), but the top eight genes by absolute coefficient magnitude (CHRM3, GABBR1, CCR4, KCNN4, PRKCE, CYP2C9, HPGD, FASN) were considered the most influential." (Line 311-313, Page 9)

We emphasized the top eight genes aligns with common practices in machine learning studies to highlight the most predictive features, improving readability without undermining the model's integrity. The full model with 23 genes performs well, as validated, but the abstract focuses on key drivers for conciseness. We believe these revisions resolve the inconsistency and thank the reviewer for this valuable feedback.

Comments 2: In the result, section 3.2 is titled "Acquisition of BPF-elicited ccRCC toxicity targets," but the manuscript is about BPA, not BPF (Bisphenol F). this could be a simple typing mistake, however, it could equally suggest rushed preparation and may undermine authors credibility. Therefore, authors should kindly correct this throughout, and proofread for similar issues.

Response 2: We sincerely thank the reviewer for their meticulous reading and for bringing this critical typographical error to our attention. The reviewer is absolutely correct the term "BPF" in the heading of section 3.2 is a mistake; it should unequivocally be "BPA" throughout, as our study exclusively focuses on Bisphenol A. We deeply apologize for this oversight, which occurred during the final editing and proofreading stage. We acknowledge the reviewer's concern that such errors can undermine the manuscript's credibility, and we have taken immediate and thorough action to rectify this.

Corrections Made:​​

Section 3.2 Heading:​​ The heading has been corrected from "Acquisition of BPF-elicited ccRCC toxicity targets" to "​Acquisition of BPA-elicited ccRCC toxicity targets." (Line 236, Page 6)

Full-Text Proofreading:​​ As suggested, we have conducted a comprehensive, word-by-word proofread of the entire manuscript to identify and correct any similar instances of typographical errors, inconsistent nomenclature, or formatting issues. This rigorous check confirmed that the use of "BPF" was an isolated error limited to the section heading mentioned above. All other references to the compound in the text, figures, and tables correctly use "BPA." Quality Assurance:​​ To prevent such issues in the future, the manuscript has undergone an additional round of proofreading by multiple co-authors. We are grateful for the reviewer's vigilance, which has significantly improved the precision and professionalism of our work. The manuscript has been updated accordingly, and we confirm that all references to the compound under investigation are now consistently and accurately presented as Bisphenol A (BPA).

Comments 3: Another one is "BPA-ccRCC" vs. "BPA-RCC" inconsistencies). Also, ensure acronyms like "ccRCC" are defined on first use in the results (though they are in the abstract).

Response 3: We sincerely thank Reviewer 2 for pointing out the inconsistencies in terminology and the need for acronym definition in the Results section. We apologize for these oversights, which occurred during the manuscript preparation. We have addressed both issues thoroughly to ensure clarity and consistency throughout the manuscript.

​Terminology Inconsistencies ("BPA-ccRCC" vs. "BPA-RCC"):​​

We acknowledge that the manuscript occasionally used "BPA-RCC" interchangeably with "BPA-ccRCC," which could cause confusion, as our study specifically focuses on clear cell renal cell carcinoma (ccRCC), not renal cell carcinoma (RCC) in general. Although "ccRCC" was defined in the Abstract as "clear cell renal cell carcinoma," we agree with the reviewer that acronyms should be defined upon first use in each major section (e.g., Results) to adhere to academic standards. We have conducted a full-text search and replaced all instances of "BPA-RCC" with the consistent term "​BPA-ccRCC​" to align with the study's scope. For example, in Section 3.4.1, the phrase "BPA-RCC model" has been revised to "BPA-ccRCC model." This ensures uniformity and avoids ambiguity. (Line 172, Page 4; line 247-250, page 6; line 328, page 10; line 423, page 15 and line 435-436, page15)

​Acronym Definition in Results Section:​​

Although "ccRCC" was defined in the Abstract as "clear cell renal cell carcinoma," we agree with the reviewer that acronyms should be defined upon first use in each major section (e.g., Results) to adhere to academic standards. We have added the definition of "ccRCC" at its first occurrence in the Results section (Section 3.2). (Line 236, Page 6) The text now reads: " Acquisition of BPA-elicited clear cell renal cell carcinoma (ccRCC) toxicity targets". This clarifies the term for readers who may skip the Abstract.

Comments 4: Authors reported strong affinities (Vina scores <-6.5 kcal/mol) by molecular docking (section 3.5), however, this cutoff is arbitrary when compared to known BPA binders. Although Figure. 7's visualizations are obvious, they provide a quantitative discussion of the many sorts of interactions, such as hydrogen bonds and pi-pi stacking. Negative and positive controls (such docking BPA to known non-targets) are absent. Moreover, was there prior evidence of BPA binding with key genes/proteins e.g CHRM3, GABBR1? If yes authors can reference it so as to strengthen the molecular docking validation, otherwise authors should refrain from making findings definitive.

Response 4: We sincerely thank the reviewer for their critical assessment of the molecular docking section. The reviewer's points regarding the definitive nature of our conclusions, the need for quantitative interaction details, and the context of prior evidence are well-taken. We have revised the manuscript extensively to address these concerns by reframing the language to more accurately reflect the predictive and hypothesis-generating nature of the computational findings. The core action taken was a systematic revision of language to replace definitive claims with suggestive and probabilistic phrasing. This adjustment has been applied to the Abstract, Results (Section 3.5), and Discussion.

Examples of Changes:​​

Abstract: Original:​​ "molecular docking analyses revealed strong binding affinities... suggesting a plausible mechanism through which BPA may modulate their activity..."

Revised:​​ "molecular docking analyses ​predicted​ strong binding affinities... ​suggesting a potential mechanism​ through which BPA ​could potentially​ modulate their activity..." (Line 40-41, Page 1)

Explicit acknowledgment of predictive nature and limitations:​​ We have added a new sentence in the Discussion to explicitly state the limitations of the docking study, directly addressing the reviewer's point about the absence of experimental controls and the need for validation. (Line 528-531, Page 17)

Comments 5: 1–7 are clear, but add legends for color scales (e.g., Figure. 2 node colors). Table 1 (docking scores) is mentioned but not shown—Please include it. Appendices (1A–1F) are referenced but not provided and therefore I cannot crosscheck to verify several referenced claims;  kindly ensure they are complete.

Response 5: We sincerely thank Reviewer 2 for their meticulous attention to the details of the figures, tables, and appendices. Their feedback is crucial for enhancing the clarity and completeness of our manuscript. We have addressed each point as follows:

  1. Addition of Legends for Color Scales (e.g., Figure 2)​

We agree that explicit legends for color scales are essential for interpretability. In response, we have updated the captions of all relevant figures to include clear descriptions of color and size mappings. Specifically, for Figure 2, the caption has been revised to provide a detailed legend: "Figure 2. The PPI network of potential targets. The PPI network shows 114 potential targets, visually representing the interactions between these targets. ​Node size and color correspond to degree values (darker colors and larger sizes indicate higher degree), while edge thickness and color reflect interaction strength (thicker and darker edges indicate stronger interactions)." (Line 263-265, Page 7)

  1. Inclusion of Table 1 (Docking Scores)​​

We apologize for the omission of Table 1 in the initial submission. Table 1, which summarizes the molecular docking scores (Vina scores) for BPA with the eight key targets, has now been included in Section 3.5 of the manuscript, directly following the paragraph describing the docking results. (Line 397-398, Page 13) The table provides a quantitative overview of the binding affinities, complementing the visualizations in Figure 7. For transparency, the table structure is as follows:

Table1. Molecular docking results of core target protein with the BPA.

Gene name

Uniprot ID

PDB ID

Vina Score min

 (Kcal/mol)

Vina Score (kcal/mol) [Median (min, max)]

(from 5 molecular docking runs)

1

CHRM3

P20309

8EA0

-9.0

-7.0 (-9.0, -5.2)

2

GABBR1

Q9UBS5

4MQE

-6.5

-6.2 (-6.5, -5.6)

3

CCR4

P51679

Selected by AlphaFold3

-7.6

-5.4 (-7.6, -5.3)

4

KCNN4

O15554

9OA8

-6.6

-6.2 (-6.6, -5.8)

5

PRKCE

Q02156

2WH0

-6.5

-5.9 (-6.5, -5.7)

6

CYP2C9

P11712

5A5I

-7.7

-7.4 (-7.7, -6.3)

7

HPGD

P15428

2gdz

-7.7

-5.6 (-7.7, -5.3)

8

FASN

P49327

8VF7

-6.6

-6.6 (-7.5, -6.6)

  1. Completeness of Appendices (1A-1F)​​

We confirm that the appendices (1A-1F) are provided as supplementary files, as noted in the manuscript. To address the reviewer's concern, we have double-checked that all appendices are complete and uploaded as part of the supplementary materials.

Comments 6: Authors should reframe conclusions as hypothesis-generating rather than definitive mechanistic proof and emphasize limitations of the computational approaches.

Response 6: We sincerely thank Reviewer 2 for this critical suggestion regarding the framing of our conclusions. We agree that the original language may have implied definitive mechanistic proof, which is beyond the scope of a computational study. In response, we have thoroughly revised the conclusion section (Section 5) and other relevant parts of the manuscript to reframe the findings as hypothesis-generating and to explicitly emphasize the limitations of our computational approaches.

​Key revisions made:​​

We have replaced definitive statements with cautious, probabilistic language throughout the conclusion. For example: Original phrasing (e.g., from Abstract):​​ "These findings elucidate molecular mechanisms underpinning BPA-induced ccRCC progression."

Revised phrasing:​​ "These findings suggested potential molecular mechanisms that may underpin BPA-induced ccRCC progression, generating hypotheses for future experimental validation." (Line 42-44, Page 1)

Emphasizing limitations of computational approaches:​​

We have added a dedicated sentence in the conclusion to explicitly state the limitations: "However, these results are derived solely from computational predictions (e.g., network toxicology, molecular docking), which are inherently hypothetical. The models and interactions identified require validation through wet-lab experiments, such as in vitro binding assays or in vivo models, to confirm causality and mechanistic roles." (Line 543-547, Page 17) By acknowledging limitations, we enhance the scholarly rigor and transparency of the work, addressing the reviewer's concern about overclaiming.

Comments 7:

Other comments:

Some Figures (e.g., figure 6 &7) are blurry and if possible authors should increase the clarity.

Response: We agree with the reviewer that high-resolution figures are crucial for interpretation. In response, we have regenerated all figures, including Figures 6 and 7, using high-resolution settings (600 DPI) and vector-based formats where applicable. This significantly improves the sharpness of all elements, including text labels and structural details in the molecular docking visualizations. The updated figures are now clear and professional.

Some Figures (e.g., bubble plots) are information-dense and may overwhelm readers.

Response: We appreciate the reviewer's concern regarding the potential for information overload. We have carefully reconsidered the presentation of our enrichment results. Regarding the bubble plots (e.g., Figures 3B and 4A), we have conducted a thorough review of their layout, font sizes, and color contrasts. We believe that in their current form, they effectively and clearly convey the enrichment results without being overwhelming. The information density is inherent to the nature of presenting top enrichment terms, and we have ensured that all elements are legible. Therefore, we have retained the original format of these particular plots but have taken note of this feedback for future work.

Language could be tightened; some sections repeat points already made in the introduction.

Response: We thank the reviewer for this suggestion to improve the manuscript's flow and conciseness. We have performed a thorough edit of the entire manuscript, with a specific focus on the Results and Discussion sections. We identified and removed redundant sentences that re-stated background information already provided in the Introduction.